# Catalytic Ozonation of Ibuprofen in Aqueous Media over Polyaniline–Derived Nitrogen Containing Carbon Nanostructures

**DOI:** 10.3390/nano12193468

**Published:** 2022-10-04

**Authors:** Angel-Vasile Nica, Elena Alina Olaru, Corina Bradu, Anca Dumitru, Sorin Marius Avramescu

**Affiliations:** 1PROTMED Research Centre, University of Bucharest, Splaiul Independenţei 91–95, Sect. 5, 050107 Bucharest, Romania; 2Department of Systems Ecology and Sustainability, Faculty of Biology, University of Bucharest, Splaiul Independenţei 91–95, 050095 Bucharest, Romania; 3Faculty of Physics, University of Bucharest, 077125 Măgurele, Romania; 4Department of Organic Chemistry, Biochemistry and Catalysis, Faculty of Chemistry, University of Bucharest, 90–92 Soseaua Panduri, 050663 Bucharest, Romania

**Keywords:** catalytic ozonation, ozone, advanced oxidation processes, water treatment, ibuprofen, polyaniline, total organic carbon, chemical oxygen demand, biochemical oxygen demand

## Abstract

Catalytic ozonation is an important water treatment method among advanced oxidation processes (AOPs). Since the first development, catalytic ozonation has been consistently improved in terms of catalysts used and the optimization of operational parameters. The aim of this work is to compare the catalytic activity of polyaniline (PANI) and thermally treated polyaniline (PANI 900) in the catalytic ozonation of ibuprofen solutions at different pH values (4, 7, and 10). Catalysts were thoroughly characterized through multiple techniques (SEM, Raman spectroscopy, XPS, pHPZC, and so on), while the oxidation process of ibuprofen solutions (100 mgL^−1^) was assessed by several analytical methods (HPLC, UV254, TOC, COD, and BOD5). The experimental data demonstrate a significant improvement in ibuprofen removal in the presence of prepared solids (20 min for PANI 900 at pH10) compared with non-catalytic processes (56 min at pH 10). Moreover, the influence of solution pH was emphasized, showing that, in the basic region, the removal rate of organic substrate is higher than in acidic or neutral range. Ozone consumption mgO_3_/mg ibuprofen was considerably reduced for catalytic processes (17.55—PANI, 11.18—PANI 900) compared with the absence of catalysts (29.64). Hence, beside the ibuprofen degradation, the catalysts used are very active in the mineralization of organic substrate and/or formation of biodegradable compounds. The best removal rate of target pollutants and oxidation by-products was achieved by PANI 900, although raw polyaniline also presents important activity in the oxidation process. Therefore, it can be stated that polyaniline-based catalysts are effective in the oxidation processes.

## 1. Introduction

Water pollution is considered a paramount issue of the modern society [1,2,3,4]. With the increase in the consumption of pharmaceuticals and personal care products (PPCPs), the presence of these emerging contaminants in water, which causes irreversible damage to human health and the ecological environment, has now become a serious environmental burden [5,6]. Owing to their bioactive activity and hazardous toxic metabolites, PPCPs are considered to have a more negative impact on the environment and water quality than any other pollutants, and appear in surface waters, groundwater, sewage water treatment plants, and industrial wastewater, at concentrations ranging from the µgL^−1^ to ngL^−1^ [5,6,7,8].

Among them, ibuprofen, as a common nonsteroidal anti-inflammatory drug, being one of the most popular, highly prescribed drug for the treatment of pain, migraine, fever, and inflammation, appears very commonly in the aqueous environment [7,9]. Approximately 90% of the therapeutic dose of ibuprofen, ranging from 600 to 1200 mg/day, is excreted in the urine and reaches the water body as metabolites or their conjugates, and 10–15% is eliminated as a free drug [6,10]. Owing to its stability and good mobility in an aquatic environment, ibuprofen has one of the highest concentrations in estuaries, which ranges between 18 and 6297 ngL^−1^ [9]. The very low concentration of PPCPs makes them difficult to remove through conventional wastewater treatment processes, with the removal rate reaching only 30–40% of drugs [5,11].

Advanced oxidation processes (AOPs) have been considered as one of the most promising alternatives for the removal of recalcitrant emerging organic pollutants from environments [3,8]. Ozonation processes such as AOPs were widely applied for degradation of refractory organic and toxic pollutants via the O_3_/H_2_O_2_ process [12,13,14]. One of the disadvantages of the single ozonation method is a slow oxidation or partial oxidation of some organic pollutants, sometimes leading to the generation of toxic intermediate products, owing to its relatively low solubility and stability in water and the selective oxidation between O_3_ and pollutants [8,13,14,15,16,17,18,19,20,21]. To overcome these problems, the catalytic ozonation process carried out with the addition of homogeneous or heterogeneous catalysts such as metal ions, metal oxidizes, and carbon-based materials has received increasing attention in recent years for efficient degradation and effective mineralization of organic pollutants [14,15]. Even if the homogenous catalytic ozonation process, which usually used metal ions as a catalyst, shows a good efficiency for the removal of organic pollutants, the presence of the metal ions might result in secondary pollution, which limited their application [14]. As a consequence, heterogeneous catalytic ozonation, which uses metal oxides, metals or metal oxides on supports, activated carbon, and minerals as catalytic systems, allows the separation of the catalyst form the solution without producing secondary pollution with solid catalysts, as well as catalyst regeneration and further utilization [14,15,22,23,24,25,26,27,28,29,30].

In recent years, carbon-based materials such as graphene, graphitic carbon nitride, mesoporous carbon, activated carbon, nanodiamonds, fullerenes, and carbon nanotubes have been successfully investigated in heterogeneous catalysis, both as supports and as catalysts, thanks to their catalytic activity and facilely-tuned surface chemistry, together with their large surface area, stability, and reusability [13,14,31,32,33,34,35,36,37]. It is now believed that the introduction of heteroatoms into carbon frameworks (e. g., nitrogen, sulfur, phosphorus, and boron) is an excellent strategy that can significantly improve the catalytic performances of carbon catalysts [32]. Among diverse heteroatoms, the introduction of nitrogen (N) into carbon frameworks has been the most widely examined for the preparation of heteroatom-doped carbon materials [38,39]. The incorporation of N into carbon frameworks modulates the electronic structures of surrounding carbon atoms to provide desirable electronic structures for many catalytic processes and can improve wetting and hydrophilic properties of the carbon material surface [32,40,41]. However, nitrogen-containing carbon nanostructures have attracted considerable attention thanks to their applicability in different scientific fields including catalysis, energy conversion/storage, and water treatment processes [42,43,44]. In this aim, the carbonization of conducting polymer nanostructures with various morphologies such as granular, tubes, fibers, and rods has the advantages of their facile conversion to nitrogen-containing carbon nanostructures that preserved the morphology of the precursors and the fact that it could facilitate the introduction of different nitrogen moieties into the carbon framework such as graphitic, pyridinic, pyrrolic, and different N-oxide species [41,45,46]. Among conducting polymers, polyaniline (PANI), which contains covalently bonded nitrogen, which remains in the carbon structure after carbonization of polymeric precursor, has been widely studied as a nitrogen-containing carbon precursor thanks to its low cost, high nitrogen content, and easy synthesis [43,44,45].

Based on the above considerations, in the present work, both PANI and nitrogen-containing carbon nanostructures obtained by carbonization of PANI were employed as catalysts in catalytic ozonation for the removal of ibuprofen. However, according to the best of the author’s knowledge, no report has been conducted so far about PANI-derived carbon used as a metal-free catalyst in catalytic ozonation of ibuprofen or other organic compounds.

## 2. Materials and Methods

### 2.1. Materials

Analytical-grade reagents were acquired from Sigma-Aldrich (Baden-Württemberg, Germany): aniline (An), ammonium persulfate (APS), sodium chloride (NaCl), ibuprofen (IBU), acetonitrile (MeCN), and trifluoroacetic acid (TFA). Airflow for ozone generation was provided by a laboratory compressor and distilled water with a resistivity of 18.2 MΩ.cm was used in all experiments.

### 2.2. Synthesis of Polyaniline

The polyaniline nanostructures were obtained by oxidation of aniline monomer (previously distilled under reduced pressure) in aqueous media in the presence of ammonium persulfate (APS) as oxidant agent without the use of acidic dopants [47]. In a typical procedure, 0.2 mL of aniline (2.0 mmol) was homogenized in 15 mL of de-ionized water using magnetic stirring at room temperature for 0.5 h and subsequently cooled in an ice bath. Then, 15 mL of APS (2.0 mmol) solution was added to the aniline water mixture. The final mixture was allowed to react at 0 °C for 12 h and precipitates were formed. The solids were collected and washed several times with de-ionized water, methanol, and ether. Finally, the product was dried in a vacuum at room temperature for 24 h and indexed as PANI. The second stage was the carbonization of PANI, and this procedure was carried out in a tubular furnace under a nitrogen atmosphere at 900 °C, with a heating rate of 3 °C/min, and the samples were kept at this temperature for 2 h. After that, the power was disconnected and the sample was left to cool to an ambient temperature under a nitrogen atmosphere. The carbonized samples are denoted as PANI-900.

### 2.3. Materials’ Characterization

The morphology of the samples was investigated by scanning electron microscopy (Carl Zeiss, Jena, Germany). The Raman spectra were measured using a Renishaw InVia Raman microscope with Ar laser excitation at 514.3 nm (Renishaw, Wotton-under-Edge, Gloucestershire, UK). The data were acquired from 2 µm diameter spots with neutral filters limiting laser power to 10%, e.g., around 0.3 mW. To avoid the sample modification, the level of the laser power was chosen after several experiments, when the power was gradually increased. X-ray photoelectron spectroscopy (XPS) analysis was performed with a SHIMADZU Kratos Axis Nova instrument (Tokyo, Japan). The spectra were excited by the monochromatized Al Kα source (1486.6 eV). Prior to individual elemental scans, a survey scan was taken for all samples in order to detect the elements present. The XPS spectra were deconvoluted with the CasaXPS software, using a weighted sum of Lorentzian and Gaussian components curves after Tougaard background subtraction. The point of zero charge (PZC) [48,49,50,51,52] was determined according to the method described by Mustafa et al. [53]. In short, a solution of 0.1 M NaCl was prepared in deionized water and 40 mL of this solution had pH adjusted to 1–12 with a 1-unit increment using HNO_3_ and NaOH solutions. In the vials corresponding to each pH value, ~0.01 g of PANI and PANI 900 was introduced and the tubes were introduced into a rotating shaker GFL 3025 (Gessellschaft für Labortechnik mbH, Burgwedel, Germany) at 20 °C. After 24 h, the pH values were recorded and the ∆pH was plotted against the initial pH. PZC is the pH value where the slope intercepts the 0X axis.

### 2.4. Experimental Set-Up for Catalytic Ozonation and Aqueous Effluents’ Analysis

All experiments regarding the oxidation of ibuprofen were performed in a semi-batch jacketed cylindrical reactor with a capacity of 250 mL equipped with magnetic stirring (Figure 1). Ozone was obtained from dried air using an ozone generator (COM-AD-01, Anseros, Tübingen, Germany) and the gas-phase ozone concentration was analyzed using an ozone analyzer BMT 964 (BMT, Stahnsdorf, Germany). The gas flow containing ozone was introduced through an inlet into the solution while stirring at 1000 rpm; the temperature was kept at 20 °C using a circulating thermostat; and solution pH was measured (PHM 240, Radiometer S.A.S., Neuilly-Plaisance, France) by an electrode inserted directly into the solution and maintained at 4, 7, or 10 according to the performed experiments by adding HCl or NaOH trough liquid inlets. Samples were withdrawn at certain time intervals (5 to 30 min) to assure complete evaluation of the oxidation process. The reactor was filled with a 200 mL aqueous solution of ibuprofen (100 mgL^−1^), 0.5 gL^−1^ catalyst, and 11 gNm^−3^ input O_3_ flow passing through it. The target pollutant concentration was determined by the HPLC method using an L-3000 system (Rigol Technologies Inc., Beijing, China) consisting of a quaternary pump, a diode-array detector, and a Kinetex C18 Evo column (150 mm × 4.6 mm i.d.; 5 µm particle size; Phenomenex, Torrance, CA, USA). The operating conditions were as follows: isocratic elution of the mobile phase composed of eluent A (0.1% TFA in water), eluent B (0.1% TFA in acetonitrile) = 20:80, flow rate 1.0 mL/min, injection volume 10 µL, detection at 220 nm, and temperature 30 °C. Oxidation by-products’ formation during the process was also evaluated by the HPLC method using the same system and a Kinetex F5 2.6 µm 100 Å, column 150 × 4.6 mm in isocratic mode (eluent 20 mM potassium phosphate, pH 1.6, flow rate: 0.5 mL/min, injection volume 10 µL, detection at 210 nm, and temperature 30 °C).

Total organic carbon (TOC) measurements were achieved by the high-temperature oxidation method using a HiPerTOC analyzer (Thermo Electron, Waltham, MA, USA). The absorbance of water samples at 254 nm (UV254) was measured using a Helios alpha (Unicam, Cambridge, UK) spectrometer. The biochemical oxygen demand (BOD) level of the final solution (after 180 min reaction) can be correlated to its content in biodegradable organic matter. The BOD measurement was achieved using a Lovibond BD600 sensor system based on the manometric, respirometric principle (manometric respirometers relate oxygen uptake to the change in pressure caused by oxygen consumption while maintaining a constant volume). The sample were diluted (1:1) with aerated water enriched with BOD seed inoculum (HACH) and nutrients (HACH-BOD Nutrient Buffer Pillows). Where necessary, a pH correction of the samples was previously performed (7.2 < pH< 7.8). The as-prepared samples were placed in bottles, connected to the BD600 sensor system, and incubated in the dark for five days at 20 °C. A blank test was conducted for each run. COD measurements were taken using a Hach COD kit and Helios alpha (Unicam, Cambridge, UK) spectrometer.

## 3. Results and Discussion

### 3.1. Materials’ Characterization

As reported, the carbonization process preserves the morphology of polymer precursors [43,44,45]. In Figure 2, the SEM image of PANI obtained by the oxidation of aniline with APS in water is presented, which shows an agglomerated nanotubular-like morphology, with different lengths of the nanotubular structure and a diameter between 200 nm and 300 nm.

Raman spectroscopy has demonstrated to be a useful analytical tool to investigate conducting polymers and carbon-based materials. Raman spectra of PANI and corresponding carbonized PANI 900 are presented in Figure 3.

Raman spectra of polyaniline show the relatively strong band at ~1594 cm^−1^, together with a shoulder at a higher frequency at ~1628 cm^−1^, attributed to C=C stretching vibration of the quinonoid ring and C–C stretching of the benzenoid ring vibrations, respectively [54,55]. The band due to the C–C stretching vibration of the quinonoid ring is observed at ~1563 cm^−1^. According to other reported results [55,56], the presence of two bands in the range between 1300 and 1400 cm^−1^ is correlated with an unequal distribution of semiquinone radical structures in chains. Thus, the band at ~1338 cm^−1^ corresponds to the C~N^+^• vibration of delocalized polaronic structures and the band at ~1390 cm^−1^ corresponds to the C~N^+^• vibration of more localized polaronic sites. The contribution due to C–N and C–H stretching vibrations in the benzenoid ring is observed at ~1245 cm^−1^ and 1180 cm^−1^, respectively. The bands observed at lower frequencies of 811 cm^−1^ and 417 cm^−1^ correspond to benzene-ring deformations and the band at ~606 cm^−1^ is indicative of ring sulfonation [54,55,56,57].

Raman spectra of PANI 900 reveal that the molecular structures of carbonized nanostructured PANIs display characteristic features of carbonaceous materials, with two broad maxima centered around ~1375 cm^−1^ and 1589 cm^−1^ [57]. The peak at a lower wavenumber, usually referred to as the disorder induced D-band, is due to the breathing vibrations of sp^2^ sites in six fold aromatic rings, and is commonly associated with structural defects and a reduction in symmetry due to the incorporation of hetero-atoms inside the graphitic lattice. The second band at higher energy, the graphitic G-band, is attributed to the stretching vibration of any pair of sp^2^ sites inside the graphitic pattern [58].

The survey scan XPS spectra of polyaniline and corresponding carbon nanostructures were taken for both samples prior to individual elemental scans (N1s) (Figure 4). All samples were calibrated using the BE of graphitic carbon at 284.5 eV. In the case of polyaniline and corresponding carbon nanostructures, the detected elements and the relative atomic concentrations are shown in Table 1.

The surface N-containing functional groups in PANI and PANI 900 were identified by deconvolution of the N1s XPS signals. N1s high-resolution spectrum of PANI was deconvoluted into four peaks at binding energies of ~398.1 eV, ~399.1, ~400, and ~401.4 eV (Figure 4a), corresponding to =N– (imine nitrogen), –NH– (benzenoid amine nitrogen), –NH^+^ (radical cation), and =NH^+^ (imine cation) functionalities, respectively [59,60]. The doping levels of the PANI nanomaterials were estimated by calculating the ratio of N+ species (sum of –NH^+^ and =NH^+^) to N species (sum of =N–, –NH–, –NH^+^, and =NH^+^). Our results show that the N^+^/N ratio, which evaluates the doping level of the polymer, is ~37% (Table 2) [59,60,61]. 

In the case of PANI 900, the chemical state of the N atom in the carbon lattice, from the N1s high-resolution spectrum, can be identified as follows: pyridinic nitrogen, pyrrolic nitrogen, quaternary nitrogen, and pyridine N-oxide [62,63]. From N1s XPS spectra of PANI 900, we identify pyridinic nitrogen at a binding energy of ~398.4 eV, pyrrolic nitrogen at ~399.6 eV, quaternary nitrogen at ~400.5 eV, and pyridine N-oxide at ~402.1 eV. Based on the contribution of the individual nitrogen type to the overall surface concentration, calculated on the basis of the area of their respective peaks, we identify that the quaternary nitrogen atoms (36.2%) are the dominant surface nitrogen species, followed by pyrrolic nitrogen (29.9), pyridinic N (25.5%), and pyridine –N– oxide (8.4%) (Table 2). Nitrogen functionalities, especially pyridinic and pyrrolic nitrogen, are electrochemically active nitrogen that can improve the charge mobility in a carbon matrix and provide a lone electron pair for conjugation with the π-conjugated rings, respectively [42].

The point of zero charge (pH_PZC_) is an important parameter with a large impact on the oxidation process because the type of charge on the catalyst surface has a significant influence on the ozone decomposition process and pollutant adsorption capacity.

From the variation in ∆pH with initial pH values, three crossings (in the acidic, neutral, and basic regions) were observed for both solids (Figure 5). This special behavior can be attributed to different sites for the acidic and basic region and a combination of them in the neutral zone. A similar type of variation was encountered in some kaolinite systems [64,65]. In the case of PANI and PANI 900, a significant number of different sites were formed, as can be seen from XPS data, and this can be attributed to emeraldine form, which corresponds to an almost 1:1 proportion of benzenoid (B) and quinonoid (Q) moieties in the polymer (Figure 6). The pH_PZC_ values are similar for both solids in the acidic and neutral region and higher for PANI 900 in the basic media, probably owing to a modification in the B/Q ratio. The ozonization process in the presence of these polymers can be favored by the conductive nature of them and to the presence of π–π systems.

### 3.2. Catalytic Ozonation of Ibuprofen in Aqueous Solutions with Polyaniline-Based Catalysts

Ibuprofen oxidation tests were performed at three pH values (4, 7, and 10), because this parameter is very important in water treatment, particularly when the catalysts’ surface presents three pH_PZC_ values. Hence, in Figure 7, Figure 8, Figure 9 and Figure 10, the variation in the ibuprofen normalized concentration along with output ozone concentration ware shown. Comparing the not-catalytic process at all three pH values (Figure 7), the strong influence of pH in the oxidation tests can be observed, which accounts for the formation of hydroxyl radicals in the basic media and is consistent with other studies [66,67,68]. At a basic pH, ozone reacts with OH^−^ to form OH• radicals according to the following reactions: O_3_ + H_2_O → 2HO• + O_2_(1)
O_3_ + OH^−^ → O_2_•^−^ + HO_2_•(2)
O_3_ + HO• → O_2_ + HO_2_• ↔ O_2_•^−^ + H^+^(3)
O_3_ + HO_2_• → 2O_2_ + HO•(4)
2HO_2_• → O_2_ + H_2_O_2_(5)

Moreover, the oxidation reactions must be correlated with pollutant pK_a_, which acts as a threshold between protonated and not protonated species. In the case of ibuprofen with pK_a_ = 4.4 (Figure 11) [69], it can be stated that, below this value, the main form of the compound is neutral and, above this value, the main form is negatively charged. Hence, in absence of catalysts, oxidation is favored at pH values ranging from 4.4 to 10, especially in the strong basic region, as it is well known that ionic forms react more rapidly and efficiently in terms of mineralization with molecular ozone.

As a result, at a basic pH, ibuprofen removal is more rapid (~56 min), compared with ~89 and ~150 min for neutral and acidic pH, respectively (Figure 7).

When the catalyst intervened, the removal of ibuprofen took place at much more rapid rates (Figure 8, Figure 9 and Figure 10): ~60 min (pH = 4), ~45 min (pH = 7), and ~29 min (pH = 10) for PANI and ~20 min (pH = 10) for PANI 900. Hence, in the presence of catalysts, the rate of target pollutant elimination is twice as high as in absence of catalysts.

Total organic carbon evolution during the oxidation process is an important tool to assess the mineralization of organic substrate. It can be observed from Figure 12 that, for all pH values, the presence of catalysts favors a thorough decomposition of ibuprofen and the most active is the system PANI 900.

The output ozone concentration is lower for a basic pH or in the presence of a catalyst, which accounts for a higher consumption in the process (Figure 7, Figure 8, Figure 9 and Figure 10). Integrating the area below the curves (Table 3) and subtracting these values from the input ozone (330 mg for 180 min), the real consumption of ozone can be obtained. In Table 3, it can be observed that the ozone consumption per mg of TOC and ibuprofen removal is considerably lower for prepared catalytic systems compared with non-catalytic processes.

Another useful parameter for monitoring the oxidation of pollutants is the absorbance of the solution at 254 nm, which provides information about the formation of aromatic compounds and further decomposition to CO_2_ and H_2_O (Figure 13). Thus, the process starts at a lower absorbance owing to ibuprofen’s low value of molar extinction coefficient and continues to increase with time until reaching a maximum, consequently followed by a decrease according to the intensity of the mineralization. Figure 13 shows that, for all pH values, the non-catalytic process presents a higher maximum of adsorption values compared with catalytic processes. Moreover, there are significant differences between the two solids used in this process, meaning that the maximum for PANI 900 reaches higher values at shorter reaction times. This accounts for a rapid decomposition of ibuprofen in aromatic compounds and, further, an advance mineralization, while for PANI, the maximum is delayed for a period of time. Similar behavior can be observed from chromatograms collected using an HPLC column with a high separation capacity (Kinetex F5 2.6 µm 100 Å, column 150 × 4.6 mm) and 210 nm detection (where the majority of organic compounds present high molecular adsorption) for 60 and 180 min reaction times (Figure 14). This qualitative approach allows an efficient comparison between oxidation processes at the three operational pH values. Thus, in the absence of catalysts, regardless of the pH zone, a significant number of oxidation by-products are formed, while in the presence of prepared solids, at both reaction times, the number and peak intensity of the formed by-products are considerably reduced.

Two other important parameters that give similar and complementary insights regarding the oxidation process and largely used in the water treatment domain are chemical oxygen demand (COD) (Figure 15) and biochemical oxygen demand (BOD) (Figure 16). From the variation in the normalized chemical oxygen demand (COD) with the reaction time, it can be seen that the oxidation process for PANI 900 has a more pronounced evolution in the degradation of the organic substrate. Concurrently, BOD values obtained for effluents at the end of the reaction time (180 min) present a variation that is consistent with the data resulting from the HPLC chromatograms (Figure 14). At pH 4, in the absence of catalysts and in spite of the presence of some residual compounds, the BOD value is near zero, which accounts for the formation of non-biodegradable by-products. In the presence of catalysts, the biodegradability of organic substrate increases, especially for PANI 900. At pH 7, the results are different because, at this value, ibuprofen is present mainly in charged form and reacts easily with molecular ozone. Hence, the concentration of biodegradable compounds increases in the absence of catalysts, while for catalytic processes, the concentration decreases owing to advanced mineralization of organic substrate. Finally, at pH 10, the level of biodegradable compounds reaches lower levels compared with pH 7 owing to more intensive degradation of the pollutants, particularly for the system PANI 900, which present the highest activity.

The reusability of the catalytic systems represents an important feature of these solids and, from Figure 17, it can be observed that, after three cycles, the activity in the oxidation process is preserved in a satisfactory manner.

### 3.3. Kinetic Study

Ibuprofen ozonation in the presence and absence of catalysts is usually modelled trough a pseudo-first-order kinetic (Equations (1)–(4)), where k_1_ and k_2_ are the reaction rates constants of organic compounds’ interaction with ozone and OH•, respectively. By plotting ln[Ibu]/[Ibu]_o_ versus time (Figure 18, Figure 19 and Figure 20), the observed constants (k_obs_ cat and k_obs not-cat_) will be obtained and are presented in Table 4. The values of the kinetic constants are used as an important tool for assessing the performances of prepared systems towards non-catalytic processes. It was emphasized that PANI 900 is more active in the acidic range, while at higher pH values, the activity became very similar.
(1)$$-\frac{d\left[\mathrm{Ibu}\right]}{\mathrm{dt}}=k_{1}\left[O_{3}\right]\left[\mathrm{Pa}\right]+k_{2}\left[\mathrm{OH}•\right]\left[\mathrm{Ibu}\right]=k_{\mathrm{obs cat}}\left[\mathrm{Ibu}\right]$$
(2)$$O_{3}+\mathrm{Catalyst}\mathrm{active}\mathrm{sites}\to \mathrm{OH}•$$
(3)$$\frac{-d\left[\mathrm{Ibu}\right]}{\mathrm{dt}}=k_{1}\left[O_{3}\right]\left[\mathrm{Ibu}\right]=k_{\mathrm{obs}\mathrm{not-cat}}\left[\mathrm{Ibu}\right]$$
(4)$$\mathrm{In}\left(\frac{\left[\mathrm{Ibu}\right]}{{\left[\mathrm{Ibu}\right]}_{0}}\right)=-k_{\mathrm{obs}}t$$


## 4. Conclusions

Catalytic ozonation is an important choice among advanced oxidation processes (AOPs) for water treatment owing to high efficiency in the removal of a large range of pollutants, including nonsteroidal anti-inflammatory drugs (NSAIDs) like ibuprofen. In this study, thermal and non-thermal treated polyaniline (PANI 900 and PANI) were used for the first time as catalysts for the oxidative destruction of ibuprofen using ozone as an oxidant agent. The presence of catalysts allows the elimination of the target pollutant twice as fast compared with non-catalytic processes, regardless of the pH of the aqueous solution. Moreover, the removal of remnant organic substrate was evaluated by various techniques (COD, TOC, BOD, UV 254, and HPLC) and it was emphasized that, in the presence of the prepared catalysts, the aqueous effluents containing ibuprofen were thoroughly purified regardless of the pH value and the high initial concentration (100 mgL^−1^). The results show that there are significant differences between the two solids, PANI and PANI 900, when there are used in the oxidation processes. Thus, PANI 900 is a more active catalyst in the degradation of ibuprofen, for all pH values, and shows a more rapid decomposition of ibuprofen in aromatic compounds and, further, an advanced mineralization of organic substrate.

## Figures and Tables

**Figure 1 nanomaterials-12-03468-f001:**
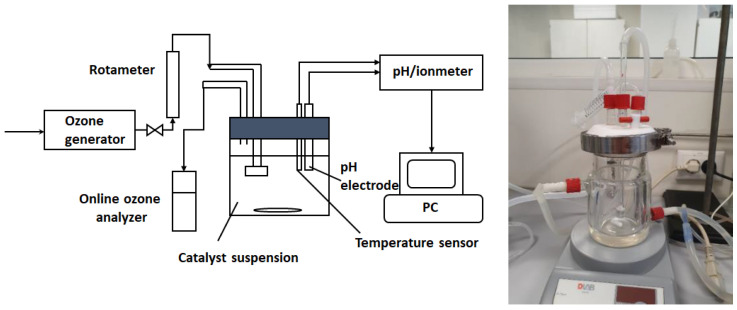
Experimental set-up used for catalytic ozonation processes.

**Figure 2 nanomaterials-12-03468-f002:**
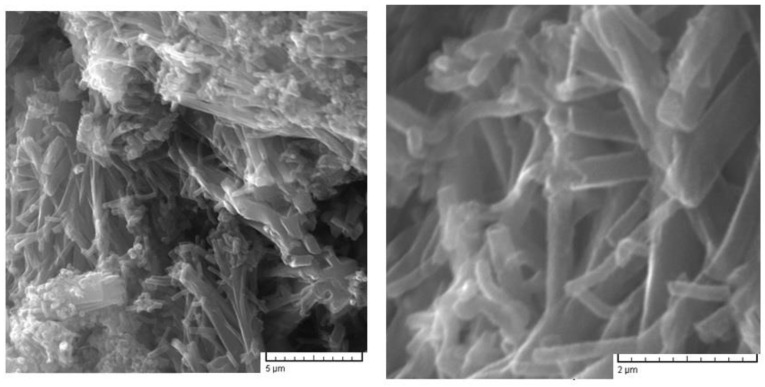
SEM image of PANI with magnitude of 8 kx (**left**) and 30 kx (**right**).

**Figure 3 nanomaterials-12-03468-f003:**
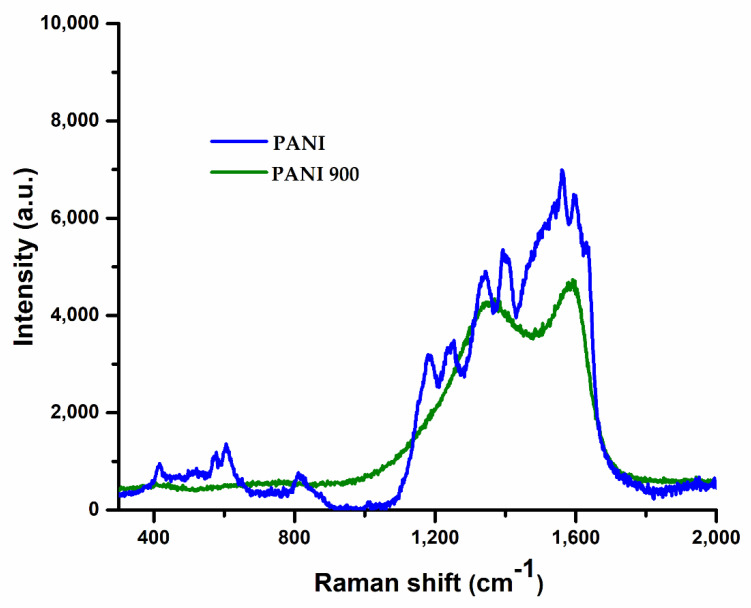
Raman spectra of PANI and PANI 900.

**Figure 4 nanomaterials-12-03468-f004:**
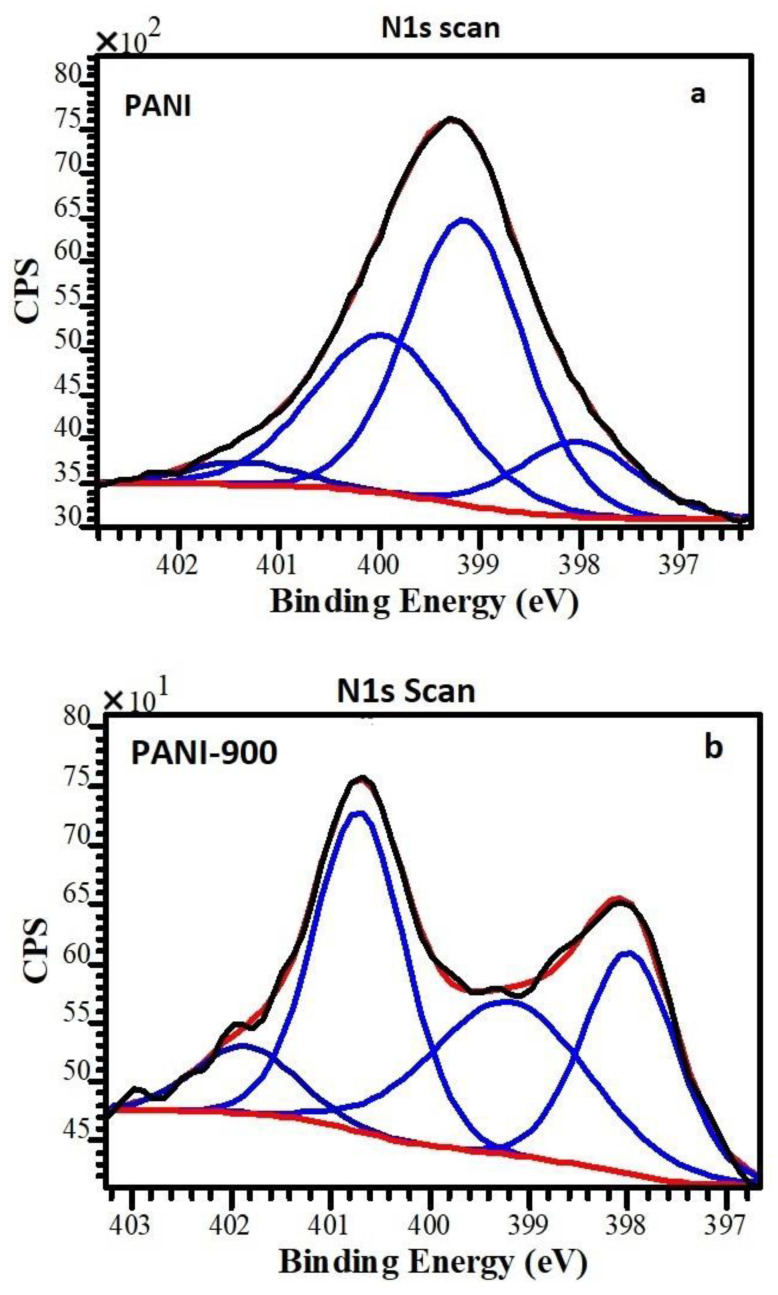
N(1s) XPS of (**a**) PANI and (**b**) PANI 900.

**Figure 5 nanomaterials-12-03468-f005:**
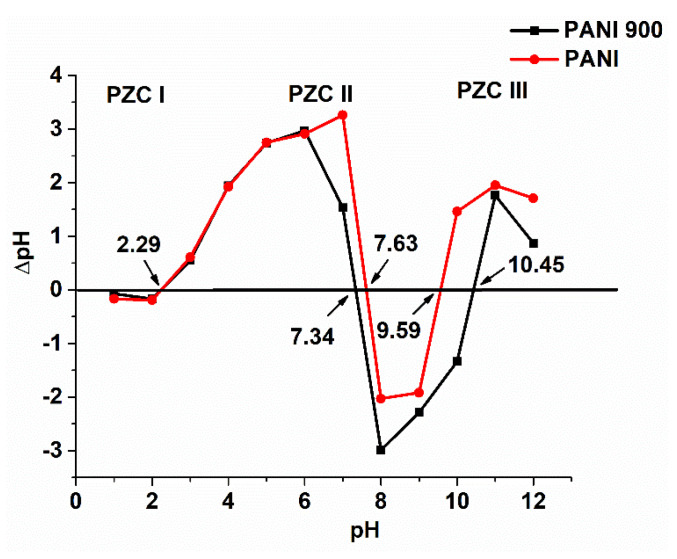
Points of zero charge for the prepared catalysts.

**Figure 6 nanomaterials-12-03468-f006:**
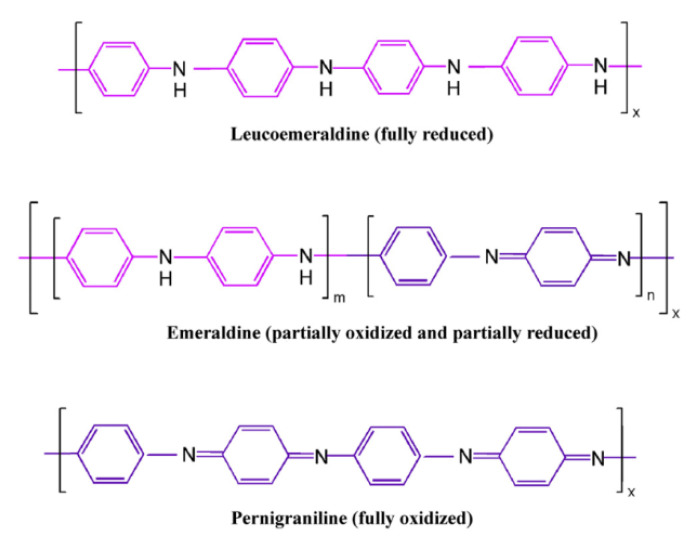
Polyaniline structures with different B/Q ratios.

**Figure 7 nanomaterials-12-03468-f007:**
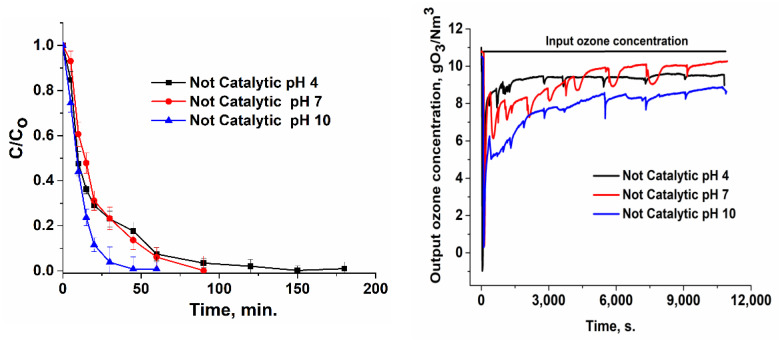
Variation in the normalized ibuprofen concentration and ozone concentration for the non-catalytic process.

**Figure 8 nanomaterials-12-03468-f008:**
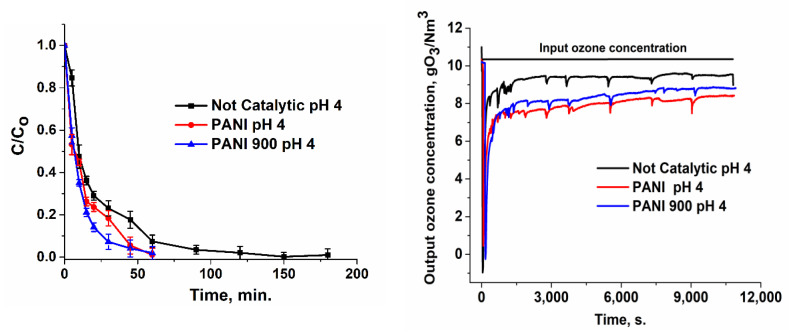
Variation in the normalized ibuprofen concentration and ozone concentration for the catalytic process at pH = 4.

**Figure 9 nanomaterials-12-03468-f009:**
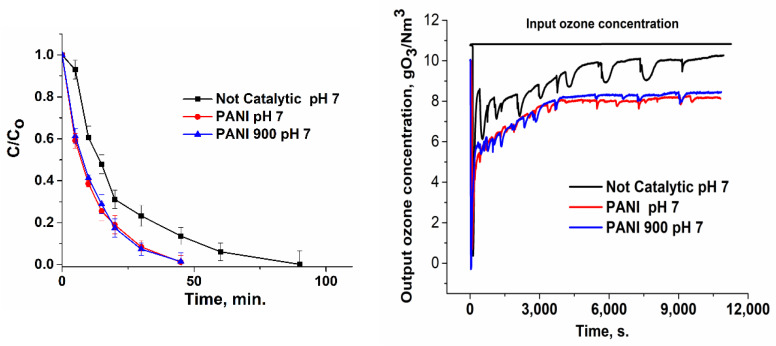
Variation in the normalized ibuprofen concentration and ozone concentration for the catalytic process at pH = 7.

**Figure 10 nanomaterials-12-03468-f010:**
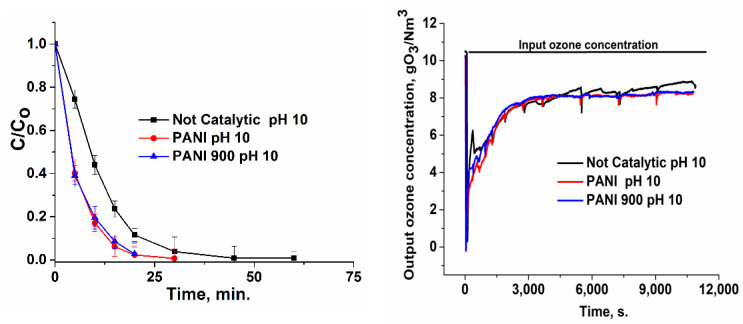
Variation in the normalized ibuprofen concentration and ozone concentration for the catalytic process at pH = 10.

**Figure 11 nanomaterials-12-03468-f011:**
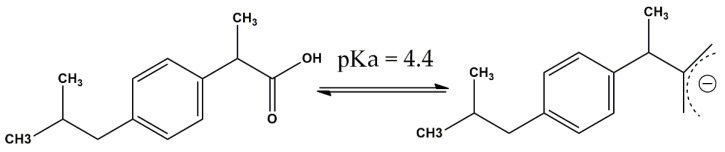
Ibuprofen structures in protonated neutral (**left**) and deprotonated form (**right**).

**Figure 12 nanomaterials-12-03468-f012:**
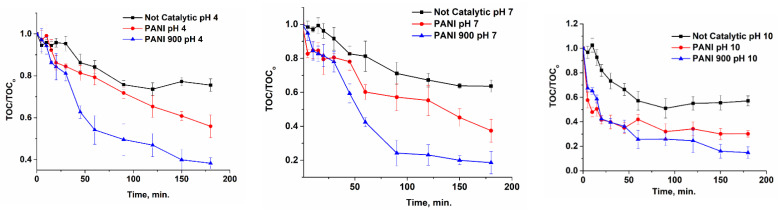
Evolution of normalized TOC for catalytic and non-catalytic oxidation processes at different pH values.

**Figure 13 nanomaterials-12-03468-f013:**
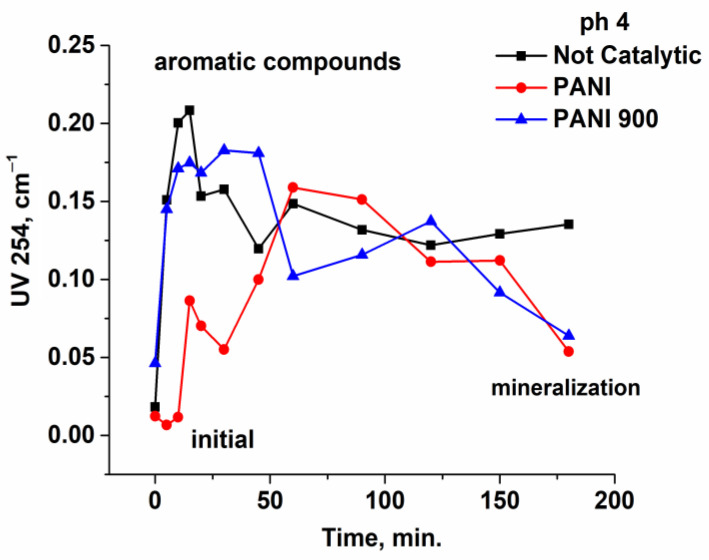
Variation in UV254 absorbance with reaction time at three pH values.

**Figure 14 nanomaterials-12-03468-f014:**
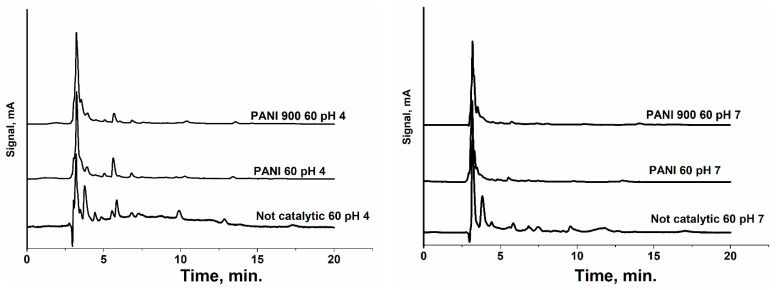
Chromatograms of samples extracted from reaction media at three pH values and different time intervals.

**Figure 15 nanomaterials-12-03468-f015:**
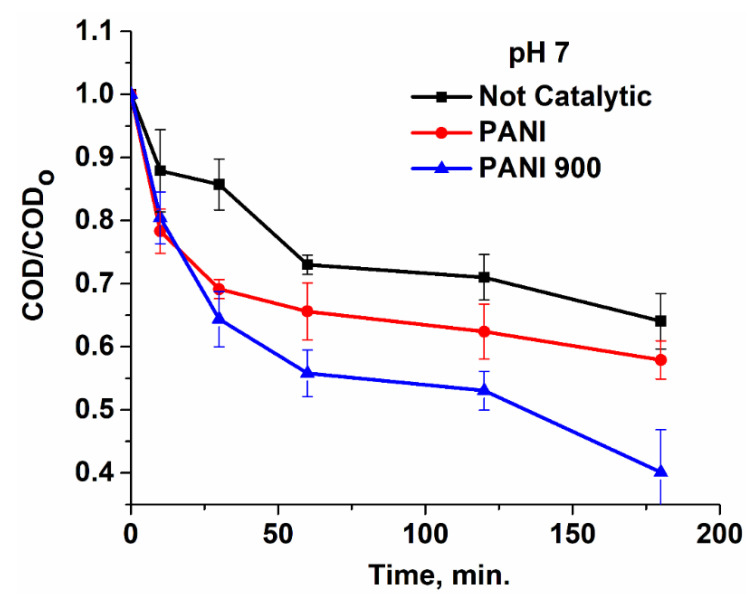
Evolution of normalized COD with reaction time in the presence and absence of catalysts.

**Figure 16 nanomaterials-12-03468-f016:**
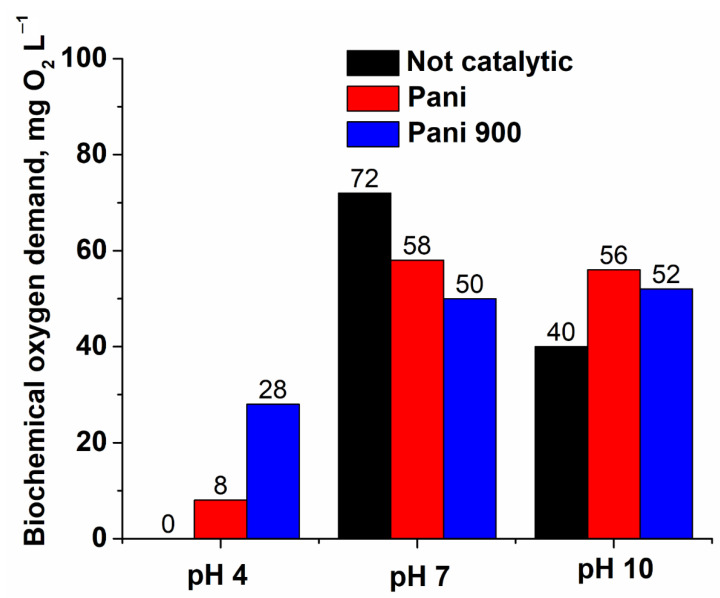
Biochemical oxygen consumption values at the end of the reaction time (180 min) for the three pH values.

**Figure 17 nanomaterials-12-03468-f017:**
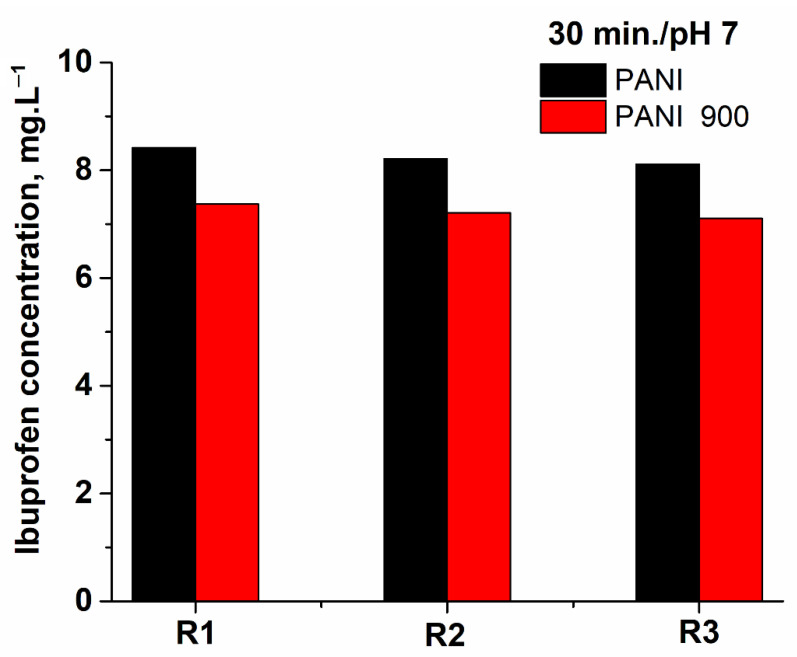
Preservation of catalytic systems’ activity after three cycles of utilization.

**Figure 18 nanomaterials-12-03468-f018:**
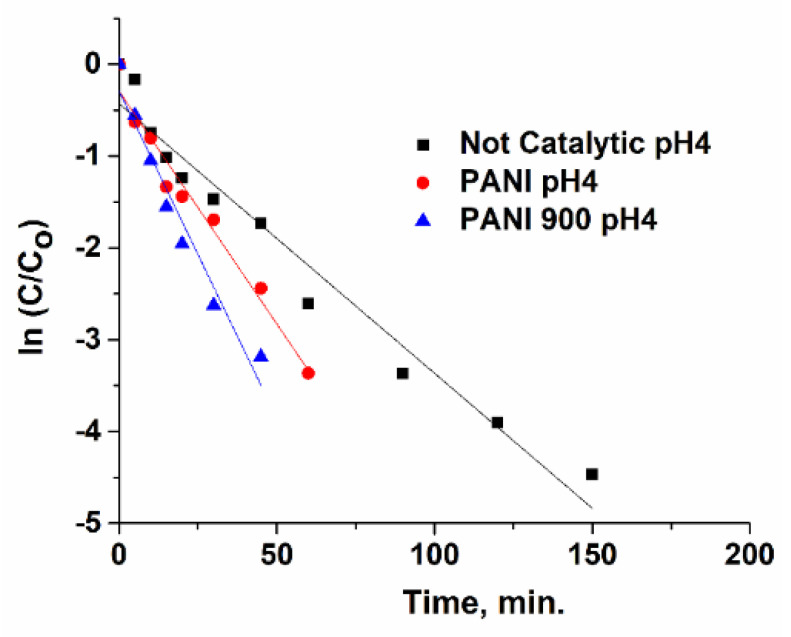
Pseudo-first-order plots of ln(C/C_o_) with reaction time for pH 4.

**Figure 19 nanomaterials-12-03468-f019:**
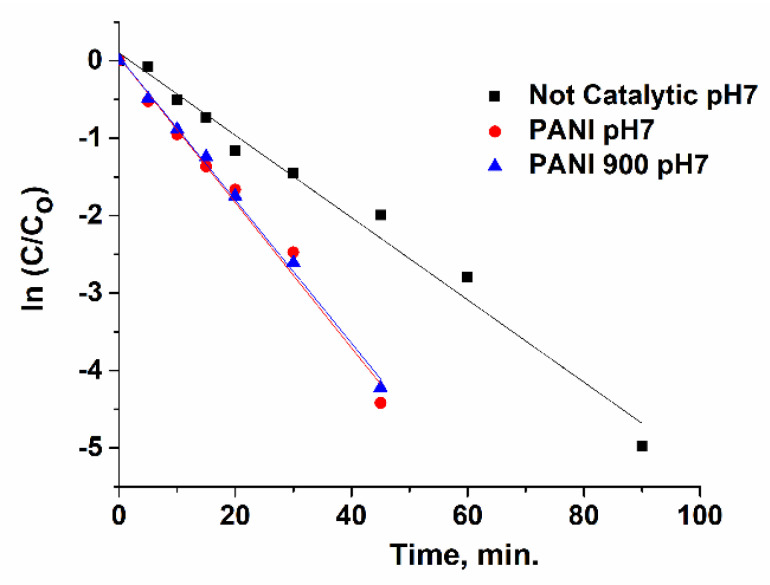
Pseudo-first-order plots of ln(C/C_o_) with reaction time for pH 7.

**Figure 20 nanomaterials-12-03468-f020:**
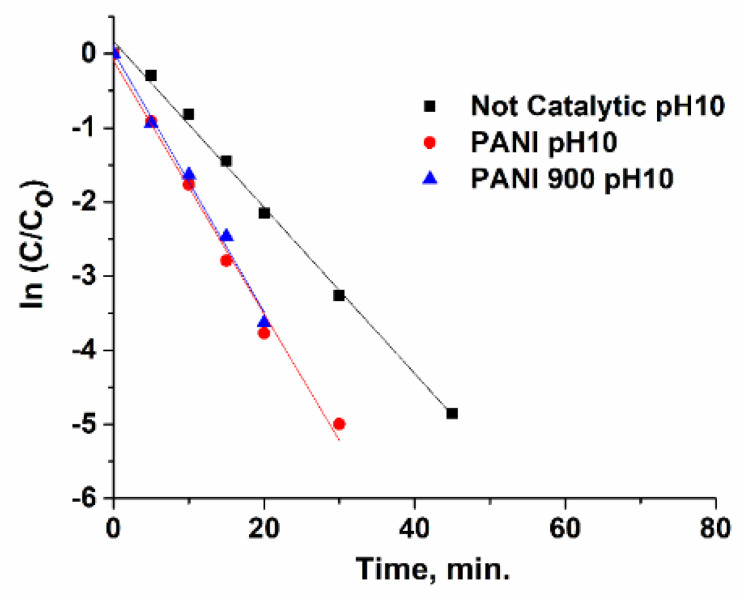
Pseudo-first-order plots of ln(C/C_o_) with reaction time for pH 10.

**Table 1 nanomaterials-12-03468-t001:** The detected elements and the relative atomic concentrations of PANI and PANI 900.

Sample	C1s		O1s		N1s		S2p	
	BE	Wt%	BE	Wt%	BE	Wt%	BE	Wt%
PANI	284.5	77.8	530.5	11.2	399.5	8.9	168.5	2.1
PANI 900	284.5	90.6	532.5	3.9	401.5	5.5	–	–

**Table 2 nanomaterials-12-03468-t002:** Relative quantities of surface-nitrogen-containing groups of PANI and PANI 900 obtained from N(1s) XPS.

Sample/N Functionality	=N–	–NH–	–NH^+^	=NH^+^	N^+^/N Ratio
PANI	17.7	45.0	30.1	7.2	0.37
Sample/N functionality	Pyridinic N	Pyrrolic N	quaternary–N	pyridine–N–oxide	Pyridinic N and Pyrrolic N
PANI 900	25.5	29.9	36.2	8.4	55.4

**Table 3 nanomaterials-12-03468-t003:** Ozone consumption during the oxidation process for prepared thin films in different conditions.

Catalytic System/pH	Total Ozone Consumed in 180 min (mg)	mgO_3_/mg TOC Removed	Time for Total Ibuprofen Removal (Minutes)	Ozone Consumed for Total Ibuprofen Removal (mg)
Non-Catalytic pH 4	52.43	3.40	180	52.43
Non-Catalytic pH 7	54.57	2.38	89	26.98
Non-Catalytic pH 10	95.26	3.52	56	29.64
PANI pH 4	93.46	3.36	60	31.15
PANI pH 7	101.77	2.58	45	25.44
PANI pH 10	105.32	2.39	30	17.55
PANI 900 pH 4	81.35	2.09	60	27.12
PANI 900 pH 7	98.39	1.92	45	24.60
PANI 900 pH 10	100.58	1.87	20	11.18

**Table 4 nanomaterials-12-03468-t004:** Kinetic parameters for the ibuprofen oxidation for different pH values.

Catalytic System/pH	k_obs_ (min^−1^) 10^2^	R^2^
Non-Catalytic pH 4	2.938	0.95869
Non-Catalytic pH 7	5.312	0.98125
Non-Catalytic pH 10	11.204	0.99521
PANI pH 4	5.067	0.96698
PANI pH 7	9.383	0.9824
PANI pH 10	17.064	0.98936
PANI 900 pH 4	7.125	0.94914
PANI 900 pH 7	9.233	0.9955
PANI 900 pH 10	17.525	0.99003

## Data Availability

All data are available upon reasonable request from the authors.

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
