# Peer review of "Catalytic Ozonation of Ibuprofen in Aqueous Media over Polyaniline–Derived Nitrogen Containing Carbon Nanostructures"

_nanomaterials, 2022, doi:10.3390/nano12193468_

Round 1
Reviewer 1 Report
In the manuscript entitled “Catalytic ozonation of ibuprofen in aqueous media over polyaniline derived nitrogen containing carbon nanostructures” the authors examined the catalytic activity of two synthesized polyaniline nanostructures (non-treated polyaniline and thermally treated polyaniline) in the catalytic ozonation of ibuprofen solutions at different pH values (4, 7, and 10). The main conclusion of the study is that the applied treatment significantly reduced the concentration of ibuprofen in aqueous effluents, and the thermally treated polyaniline is more active in the degradation of ibuprofen than the non-treated polyaniline.
Removal of drugs and their metabolites from surface waters and wastewater is very important as they can pose a serious threat to the environment and human health. This means that the research presented in the manuscript is topical and could be of interest to readers of the Nanomaterials.
The manuscript is correctly organized. The title reflects the article’s content. The experimental details and analytical procedures are well described. The results are clearly presented. However, there are several points that must be improved or clarified before publication.
- In my opinion, the catalyst dose used by the authors in the experiment (0.5 g/L) was quite high. Please explain, why such a dose of catalyst was used? Have the authors tested the effect of different doses of catalysts on ibuprofen removal?
- Why did the authors choose the ibuprofen concentration of 100 mg/L for the study, when in the Introduction the authors wrote that the ibuprofen concentration in the estuaries is in the range of 18-6297 ng/L.
- Figures 8, 9 and 10 show that the output ozone concentration, especially for the catalyst trials, stabilizes at about 8 g O3/Nm3. This is about 80% of the ozone input concentration, so even after 180 minutes, the ozone consumption is still about 20% of the ozone supplied. Please explain this phenomenon, especially since, the normalized concentration of ibuprofen is close to 0 after about 50 minutes.
- The manuscript contains linguistic and editing errors, e.g.,
· Lines 13 and 14: it should be “thermally” instead of “thermaly”,
· Line 335: “in” is written twice,
· Line 339: at 210, a unit is missing,
· Figure 10 , Table 3, Figure 13 and Table 4: it should be “pH” instead of “ph”,
· Figure 16, Y-axis: it should be “Biochemical oxygen demand” instead of “Biological oxigen demand”.
Please read the article carefully and correct any mistakes.
Overall, the manuscript presents quite interesting results and is worth to consider for publication in the Nanomaterials.
Author Response
Dear reviewer,
Thank you for your kind and useful suggestions related to the content of our paper. Al your requirements were answered and I think that in the new form of the paper a better form.
Lines 13 and 14: it should be “thermally” instead of “thermaly”, · Line 335: “in” is written twice, · Line 339: at 210, a unit is missing, · Figure 10, Table 3, Figure 13 and Table 4: it should be “pH” instead of “ph”, · Figure 16, Y-axis: it should be “Biochemical oxygen demand” instead of “Biological oxigen demand”.
These errors were corrected
In my opinion, the catalyst dose used by the authors in the experiment (0.5 g/L) was quite high. Please explain, why such a dose of catalyst was used? Have the authors tested the effect of different doses of catalysts on ibuprofen removal? Why did the authors choose the ibuprofen concentration of 100 mg/L for the study, when in the Introduction the authors wrote that the ibuprofen concentration in the estuaries is in the range of 18-6297 ng/L. Figures 8, 9 and 10 show that the output ozone concentration, especially for the catalyst trials, stabilizes at about 8 g O3/Nm3. This is about 80% of the ozone input concentration, so even after 180 minutes, the ozone consumption is still about 20% of the ozone supplied. Please explain this phenomenon, especially since, the normalized concentration of ibuprofen is close to 0 after about 50 minutes.
• In this type of studies the used catalyst dose encompass a large range according to the volume of solution and concentration of pollutant. Thus there are encountered even higher doses in different studies: 1-2 g/L for COD =1220 mg/L and turbidity 247 NTU (https://doi.org/10.1016/j.jwpe.2020.101597); 0.5 g/L (http://dx.doi.org/10.1016/j.watres.2015.05.052), evolution from 0 to 10 g/L; (10.1016/j.jhazmat.2008.05.094) and a study that I co-authored (first author) 2g/L (10.1007/s10562-008-9812-y) etc. There are also studies with dose under 0.5g/L: 10.1016/j.apcatb.2008.07.007; https://doi.org/10.1016/j.cej.2019.03.192 etc. The rationale for selecting 0.5 g/L was to better emphasize the difference between the non-catalytic and catalytic processes considering the target pollutant initial concentration and the persistence of organic substrate until mineralization. I tested in other papers the elimination of various pollutants at different catalyst dose and corroborated with literature data I choose 0.5 g/L as an optimum dose. I find more important to assess the influence of pH considering the 3 values of pHPZC.
• I consider to use as initial concentration of ibuprofen 100 mg/L to better observe the differences between presence and absence of catalysts, between catalysts (were the differences are harder to be observed) and to assess the activity of these catalysts at higher concentrations. Also, during the oxidation process there are some resistant to oxidation by-products and elimination of them reflect to a greater extent the efficiency of the catalysts.
• The relative high flow rate of gaseous effluent (10 L/h) and the low solubility of O3 in water are responsible for a plateau in the ozone concentration output at relative higher concentration. The solution for that is a recirculation of a fraction of the gaseous effluent. This will be a subject for further studies.
Reviewer 2 Report
This work is to compare the catalytic activity of polyaniline (PANI) and thermaly treated polyaniline (PANI 900) in the catalytic ozonation of ibuprofen solutions at different pH values. The remnant organic substances during the degradation of ibuprofen have also been carefully evaluated by various techniques such as COD, TOC, BOD, UV 254, and HPLC. The experiments were comprehensive, and the data was abundant. However, there still are some details need to be modified and some questions need to be answered.
1. The clarity of Figure 8 should be improved
2. The writing method in the manuscript should be unified, such as pH and ph in Figure 10.
3. Strangely, it can be seen from Figure 10 that there is a significant difference in the degradation of ibuprofen under the condition of pH=10 between the system with and without the catalyst, why does Table 3 show the difference of ''total ozone consumed in 180 min'' values are only in the 10 mg range.
4. Line 360 ''CCOD''?
5. Line 385, ''Kinetic study'' is missing the title number
6. The author should explain that the catalytic activity of PANI900 is higher than that of PANI, but from the results in Table 4, the performance of the two materials is relatively close, especially when the pH=7, the removal of ibuprofen by PANI is higher.
Author Response
Ref 2
Dear reviewer,
Thank you for your kind and useful suggestions related to the content of our paper. Al your requirements were answered and I think that in the new form of the paper a better form.
This work is to compare the catalytic activity of polyaniline (PANI) and thermaly treated polyaniline (PANI 900) in the catalytic ozonation of ibuprofen solutions at different pH values. The remnant organic substances during the degradation of ibuprofen have also been carefully evaluated by various techniques such as COD, TOC, BOD, UV 254, and HPLC. The experiments were comprehensive, and the data was abundant. However, there still are some details need to be modified and some questions need to be answered.
- The clarity of Figure 8 should be improved
The figure was improved
- The writing method in the manuscript should be unified, such as pH and ph in Figure 10.
The errors were corrected.
- Strangely, it can be seen from Figure 10 that there is a significant difference in the degradation of ibuprofen under the condition of pH=10 between the system with and without the catalyst, why does Table 3 show the difference of ''total ozone consumed in 180 min'' values are only in the 10 mg range.
Your observation is correct. However, the total ozone consumed in 180 was mainly used for differentiating the non-catalytic and catalytic processes. For catalyst activity assessment the other columns are more representative (e.g. mgO3/mg TOC removed).
- Line 360 ''CCOD''?
The error was corrected.
- Line 385, ''Kinetic study'' is missing the title number
The error was corrected.
- The author should explain that the catalytic activity of PANI900 is higher than that of PANI, but from the results in Table 4, the performance of the two materials is relatively close, especially when the pH=7, the removal of ibuprofen by PANI is higher.
Again your observation is correct. However, kobs is obtained from ln( [Ibu]/[Ibu]0) and in this situations the errors are higher comparing with C/C0 plots. Again the kinetic plots are meant for emphasize the differences between the absence and presence of catalysts.
Reviewer 3 Report
In this article, the authors present a comparative study on the catalytic activities of polyaniline and heat-treated polyaniline in the catalytic ozonation of ibuprofen solutions. The synthesis of the catalysts is described; some techniques are used to characterize the catalysts, including SEM, Raman and XPS, while activity tests are conducted at three different pH values. Experimental data demonstrate the activity of catalysts in the degradation of ibuprofen, and the importance of heat treatment on polyaniline.
In my opinion, this study presents a sufficient degree of novelty, and can be proposed for publication in this Journal. However, I suggest some changes to improve the quality of the manuscript writing.
1. Abstract does not conform to standards. It looks like an introduction rather than an abstract. An abstract consists of four basic parts and can be considered an excerpt from the article. The classic scheme is: a brief introduction to the problem to be addressed, methods in brief, a summary of the main results and a brief conclusion. All this in about ten lines, then 2/3 lines on each part.
2. Figure 2. The SEM, magnitude is missing. A comment on the SEM is also missing. What is the SEM for the purpose of the job? It is necessary to comment on the SEM and correlate to the results. In general, the results of the characterization should be related to the catalytic activity, to give consistency.
3. Figures 7-12 must be homogeneous. Same font size. Is the abbreviation for second sec or s? I believe s. The scale must always be the same, choose between minutes and seconds.
4. Figures 13 and 14. What is the unit of measurement in ordinate?
5. Bibliographic references are redundant, and the style does not conform to that suggested by the Journal.
Author Response
Dear reviewer,
Thank you for your kind and useful suggestions related to the content of our paper. Al your requirements were answered and I think that in the new form of the paper a better form.
In this article, the authors present a comparative study on the catalytic activities of polyaniline and heat-treated polyaniline in the catalytic ozonation of ibuprofen solutions. The synthesis of the catalysts is described; some techniques are used to characterize the catalysts, including SEM, Raman and XPS, while activity tests are conducted at three different pH values. Experimental data demonstrate the activity of catalysts in the degradation of ibuprofen, and the importance of heat treatment on polyaniline.
In my opinion, this study presents a sufficient degree of novelty, and can be proposed for publication in this Journal. However, I suggest some changes to improve the quality of the manuscript writing.
- Abstract does not conform to standards. It looks like an introduction rather than an abstract. An abstract consists of four basic parts and can be considered an excerpt from the article. The classic scheme is: a brief introduction to the problem to be addressed, methods in brief, a summary of the main results and a brief conclusion. All this in about ten lines, then 2/3 lines on each part.
The abstract was corrected
- Figure 2. The SEM, magnitude is missing. A comment on the SEM is also missing. What is the SEM for the purpose of the job? It is necessary to comment on the SEM and correlate to the results. In general, the results of the characterization should be related to the catalytic activity, to give consistency.
The SEM, magnitude was added. We choose to introduce in the manuscript the SEM image due to the synthesis method used for polyaniline. According with the synthesis method, it was supposed to obtain the tubular like morphology for polyaniline that was the meaning of the SEM image in our manuscript. The morphology of the catalyst can have an influence on ozone interaction with catalyst surface and in numerous studies this type of morphology is used and intentionally obtained due to increasing the accessibility of pollutants and ozone to the surface.
- Figures 7-12 must be homogeneous. Same font size. Is the abbreviation for second sec or s? I believe s. The scale must always be the same, choose between minutes and seconds.
The figures were corrected. Abbreviation is s. not sec. and was corrected. I preserve the minutes for Ibuprofen variation and seconds for ozone variation since the online ozone analyzer measure at every 10 seconds the O3 concentration and the aqueous samples were withdrawn from the reaction at different times measured in minutes. So it is difficult to change one of them in the other measurement unit. Please accept that.
- Figures 13 and 14. What is the unit of measurement in ordinate?
The UV absorbance is dimensionless and usually the optical path for cuvettes is 1 cm. However, there is the possibility to use different cuvettes hence the better way is to use cm-1 as measurement unit for UV 254. For chromatograms we use signal measured in mA as resulted from HPLC software.
- Bibliographic references are redundant, and the style does not conform to that suggested by the Journal.
Since the catalytic ozonation is a complex process in terms of materials preparation and characterization or oxidation and we need to use references from different sources in order to distinguish our work from others and to pinpoint the novel findings. The reference style was changed with MDPI approved form.